# Mapping Climate Change, Natural Hazards and Tokyo's Built Heritage

**Peter Brimblecombe [1,2,3,*], Mikiko Hayashi [4] and Yoko Futagami [5]**

[1] School of Energy and Environment, City University of Hong Kong, Tat Chee Avenue, Kowloon, Hong Kong

[2] Department of Marine Environment and Engineering, National Sun Yat-Sen University, Kaohsiung 80424, Taiwan

[3] Aerosol Science Research Center, National Sun Yat-Sen University, Kaohsiung 80424, Taiwan

[4] Center for Conservation Science, National Institutes for Cultural Heritage Tokyo National Research Institute for Cultural Properties, Tokyo 110-8713, Japan; hayashi03@tobunken.go.jp

[5] Department of Art Research, Archives and Information Systems, Tokyo National Research Institute for Cultural Properties, Tokyo 110-8713, Japan; futa@tobunken.go.jp

* Correspondence: p.brimblecombe@uea.ac.uk

**Abstract:** Although climate change is well recognised as an important issue in Japan, there has been little interest from scientists or the public on the potential threat it poses to heritage. The present study maps the impact of emerging pressures on museums and historic buildings in the Tokyo Area. We examine a context to the threat in terms of fluctuating levels of visitors as a response to environmental issues, from SARS and COVID-19, through to earthquakes. GIS mapping allows a range of natural and human-induced hazards to be expressed as the spatial spread of risk. Temperature is increasing and Tokyo has a heat island which makes the city hotter than its surroundings. This adds to the effects of climate change. Temperature increases and a decline in relative humidity alter the potential for mould growth and change insect life cycles. The region is vulnerable to sea level rise, but flooding is also a likely outcome of increasingly intense falls of rain, especially during typhoons. Reclamation has raised the risk of liquefaction during earthquakes that are relatively frequent in Japan. Earthquakes cause structural damage and fires after the rupture of gas pipelines and collapse of electricity pylons. Fires from lightning strikes might also increase in a future Tokyo. These are especially relevant, as many Japanese heritage sites use wood for building materials. In parallel, more natural landscapes of the region are also affected by a changing climate. The shifting seasons already mean the earlier arrival of the cherry blossom and a later arrival of autumn colours and a lack of winter snow. The mapping exercise should highlight the spatial distribution of risk and the way it is likely to change, so it can contribute to longer term heritage management plans.

**Keywords:** earthquakes; fire; floods; historic sites; landslides; museums; insects; sea level rise; typhoons; visitors

## 1. Introduction

Our heritage is under threat. The need to protect Japanese tangible heritage from disasters has been addressed by government funding increases: JPY 2905 million in 2019, to JPY 3907 million in 2020. While natural hazards are well recognised issues in Japan and climate science is strong, there has been relatively little interest there from scientists or the public on the potential threat a changing climate poses on heritage, especially in the way they alter the frequency of

meteorologically driven hazards. The present study maps the impact of external pressures on museums, historic buildings such as temples and shrines in the Tokyo Area. This is one of the most populous metropolitan areas in the world, which includes several prefectures of the Kantō Region of Japan, as well as Yamanashi Prefecture. The Tokyo Metropolis is elongated from east to west, stretching from mountains which stand in the west to Tokyo Bay to islands scattered over the Pacific Ocean; although in this study islands are excluded, yet it still covers an area of some 1790 square kilometres. The *Japan Meteorological Agency* (JMA) places Tokyo in the EJP climate zone (Pacific side of eastern Japan), with hot and humid summers and cold winters. In winter, the wind from the Siberian continent causes heavy snow on the Sea of Japan side, but when the air crosses the mountains to Tokyo and reaches the Pacific side to Tokyo, it becomes dry air. The winter is the driest season, thus the season is sunny and fairly mild, but the city experiences hot, humid and rainy summers.

Japan is an island country in East Asia (Figure 1a) and Tokyo, excluding islands, is located between 35° 30′ 05″ to 35° 53′ 54″ North latitude and 138° 56′ 35″ to 139° 55′ 07″ East Longitude (Figure 2a). The topography gradually decreases in elevation from the western mountains to the alluvial lowlands, ending at Tokyo Bay (Figure 1c). The terrain can be regarded as defined by the Tamagawa River catchment, and the geographical characteristics are completely different between east and west, with the boundary near Ome City, the estuary settlement in the Kantō Mountains. There are accurate 5 m-mesh data as a quantitative representation of terrain within the Digital Elevation Model (DEM) provided by the Geospatial Information Authority of Japan [1].

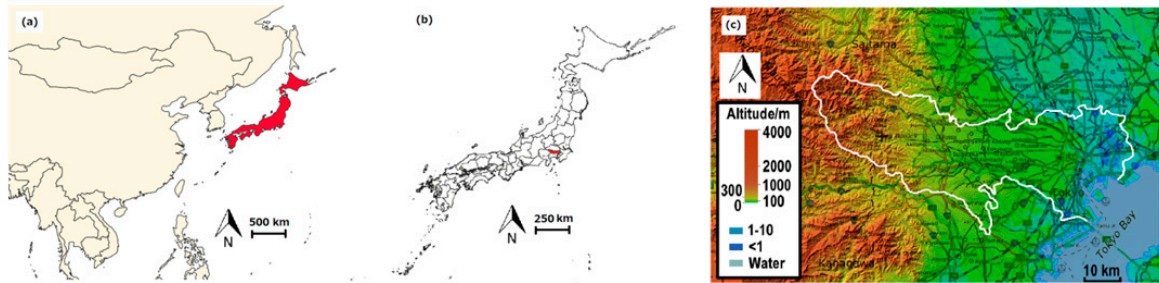

**Figure 1.** (**a**) Japan's position in East Asia. (**b**) Tokyo's position in Japan. Source: National Land Numerical Information (https://nlftp.mlit.go.jp/[2] (**c**) Topography of the Tokyo region discussed in this paper, with a white line denoting the boundary of the Tokyo Area.

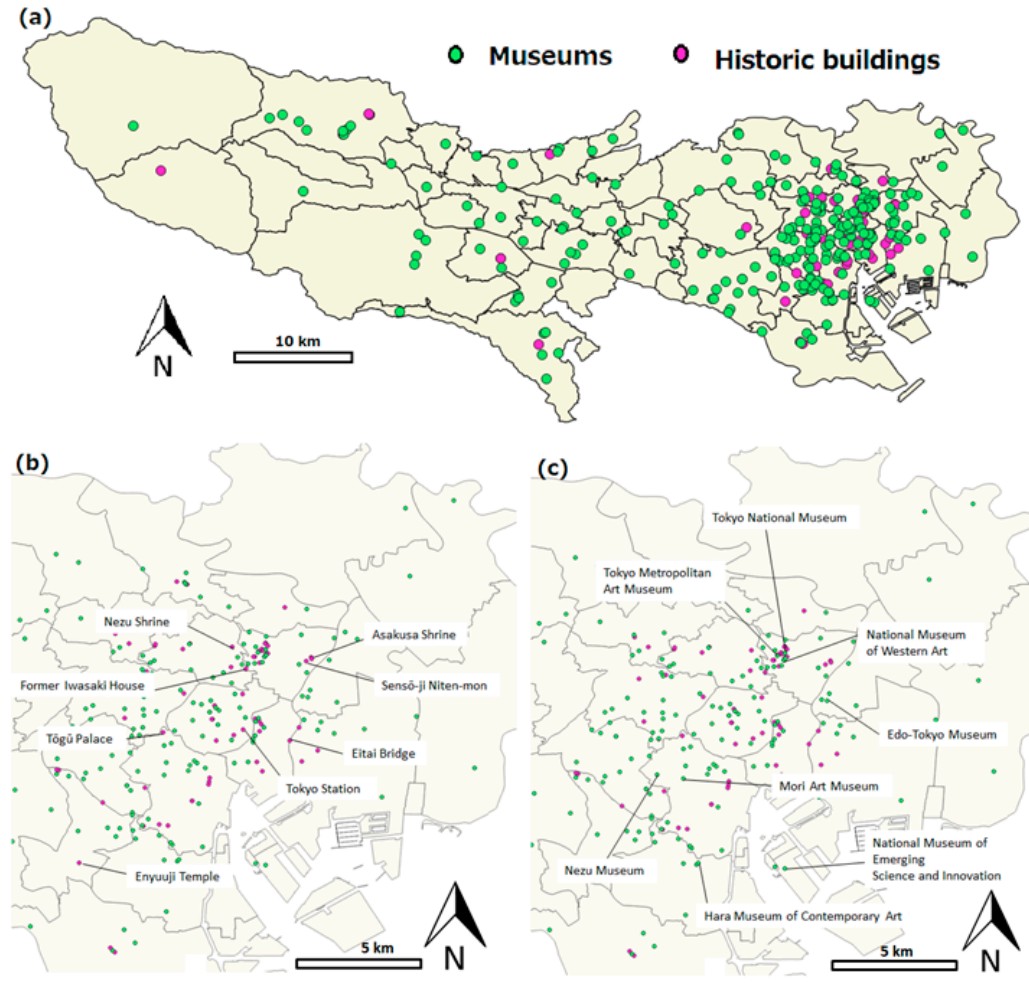

**Figure 2.** Location of (**a**) Historic buildings (pink) and museums (green) in Tokyo, (**b**) major historic buildings and (**c**) museums in central Tokyo. Note: Source: National Land Numerical Information (https://nlftp.mlit.go.jp/ [2]).

Tokyo, as a major city with a lengthy history, has a great wealth of heritage. There are numerous historic buildings and sites shrines, temples and great buildings, perhaps some favourites being: (i) Asakusa Shrine, (ii) Tōgū Palace, (iii) Tokyo Station, (iv) Sensō-ji Niten-mon, (v) Nezu Shrine, (vi) Enyuu-ji Temple, (vii) Former Iwasaki House, (viii) Eitai Bridge, (ix) Shofuku-ji Temple Jizōdō. Add to this many historic buildings; these marked as 83 points in pink (Figure 2a,b) and all of them are designated buildings and structures as National Treasure or Important Cultural Properties. Tokyo has hundreds of museums and galleries, some of the favourites being: (i) Tokyo National Museum, (ii) Edo-Tokyo Museum, (iii) Ghibli Museum, (iv) Nezu Museum, (v) Hara Museum of Contemporary Art, (vi) Mori Art Museum, (vii) Tokyo Metropolitan Art Museum, (viii) National Museum of Western Art, and (ix) the National Museum of Emerging Science and Innovation. Museums are marked as 232 points in green (Figure 2a,c). Maps of heritage and the risk imposed by climate change are often seen as important tools for the strategic management of heritage, e.g., [3,4], with some at a national or regional level [5–8]. Among a few examples of Japanese studies in this field, [9,10] integrated a database of the nationally designated cultural properties and a map of the active faults with GIS to estimate the seismic risk of each property. Difficulties in such mapping involve the problem of scaling because data are typically collected at national or regional levels rather than at city scale. It is also problematic to tune the data, often collected for other purposes, to heritage; e.g., seismic risk is defined for buildings in general, not

heritage, or that meteorological information is collected for many purposes, so the risks imposed on heritage requires considering the notion of heritage climate [3,11].

Special threats to the region are flooding and the increased pressures that climate change may present in terms of both river and sea floods [12,13] and the failure of sea defences [14]. As Tokyo is a coastal city, sea level rise makes low lying areas additionally vulnerable to increased flooding, with Estaban et al. [15], arguing that "The combined effect of an increase in typhoon intensity and sea level rise could pose significant challenges to coastal defences around Tokyo Bar around the turn of the twenty-first century." There are also risks from earthquakes, which are regularly mapped in the city as this affects a range of issues in addition to the threat to built heritage, e.g., real estate prices [16]. The Bureau of City Planning, Tokyo Metropolitan Government maps earthquake threat in their regional risk measurement survey on earthquakes, but it is also the subject of damage forecasts. Although air pollution and acid rain represent threats, they are not treated in this paper. Tokyo has worked hard to improve its air quality [17] and the study of acid rain has a long history in Japan [18], so despite concerns about the high level of threat especially from Chinese emissions [19] these have been much reduced in the current decade [20].

## 2. Materials and Methods

This study uses data on climate and natural hazards relevant to looking at long term pressures on built heritage in the Tokyo area. In addition to a range of research articles, we make use of government and municipal reports to assess the magnitude of the threat from a changing environment. A range of climate projections available from the Japanese Meteorological Agency [21] are especially useful in estimating likely future change. Geographic information system (GIS) allows the threat to be mapped.

We used the software QGIS, originally *Quantum GIS*, which allows users to analyse and edit spatial information in multiple raster formats and as vector data; which can be stored as either point, line, or polygon features. We particularly adopted 3.6 Noosa developed by the QGIS community. The software provides viewing, editing and analysis capability to overlap a range of natural hazards in the area and their changes over time. Vector data of points and areas, such as the location of built heritage (historical sources and museums), maps of Tokyo, the area of inundation by flooding and area of potential sediment disasters, were obtained from the National Land Numerical Information Download Service [2]. Areas of inundation are calculated for each river under designated storm impacts. Where vector data of point and area data were not available, we have resorted to simple counts from manually overlapped data.

## 3. Results

Pressures on our heritage are much discussed, and the impact of external local and global events is clear from Figure 3, where the numbers of foreign visitors to Japan are plotted. Pressures from epidemics, financial collapse and geophysical events cause fluctuating visitor numbers. The most recent crisis of COVID-19 has had a special impact on the heritage sector, where it has greatly affected visitor numbers and the attendant loss of income, though in some cases the most fragile sites experienced a welcome respite from heavy flows, allowing natural sites to recover a little [22]. There will also be problems with the subsequent cleaning of the viral contamination from properties and interior surfaces.

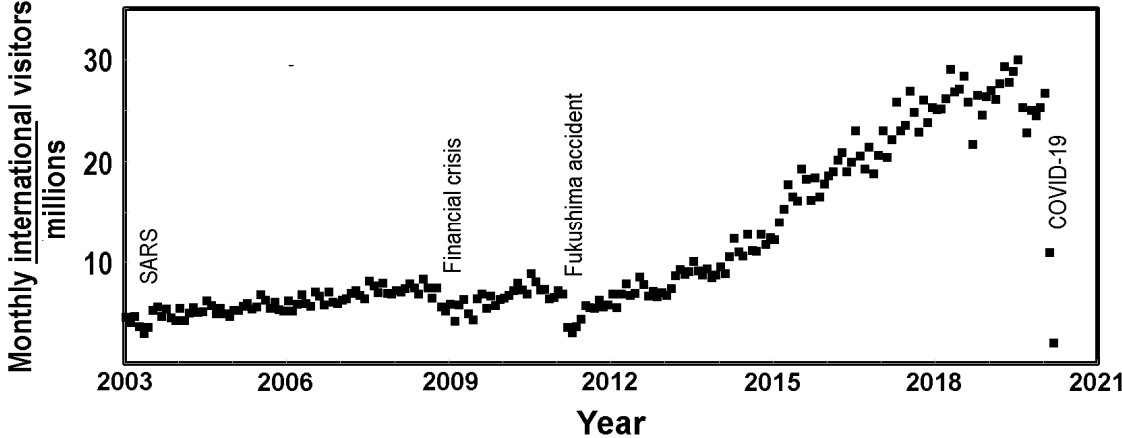

**Figure 3.** International visitors to Japan from Japan National Tourism Organization.

### 3.1. Temperature and Urban Heat Island

Temperature is the best understood parameter under a changing climate; likely to be affected through the addition of radiatively active gases to the atmosphere. Globally, the temperature is likely to have increased by about 0.8 °C through the 20th century. The Intergovernmental Panel on Climate Change (IPCC) *Fifth Assessment Report* suggests that surface temperature will rise a further 0.3 to 1.7 °C during the current century. In Japan from 1931 to 2017, the average temperature in Tokyo increased at a rate of 3.2 °C/century [23]. Projecting the regional climate of Eastern Japan between the periods 1980–1999 and 2076–2095 suggests that the annual mean temperature in Tokyo, where there is a strong heat island, will increase by more than 3 °C in almost a century.

In recent years, high temperatures in summer have been noted in large cities such as Tokyo, and public interest has been engaged by the added contribution the urban heat island makes to the increase in the number of heat stroke patients. The annual mean temperature in central Tokyo has increased about 3.2 °C/century, which is 4.4 times faster than the global mean temperature (0.73 °C/century) and 2.1 times faster than the Japanese mean temperature of 1.21 °C/century (JMA, 2019). These increases are partly due to the development of the urban heat island. The current trends for this effect, studied by Lee et al. [24], suggest that the summer heat island in Tokyo is increasing at 0.85 °C/century, while Manila, Seoul and Mumbai were slower, at 0.15, 0.2 and 0.36 °C/century, respectively. The heat island clearly substantially adds to the effect of greenhouse warming. The speed of temperature change in Tokyo has drawn popular comment [25] and the urban heat island is generally seen to arise from the decrease in green and water areas, the increase in ground covered with asphalt and concrete, the increase in heat (exhaust heat) generated from cars and buildings and the poor ventilation by wind in the street canyons [26]. Rising temperatures in summer have been a particular problem, because the heat island has helped reduced the comfort of urban life and affected human health. Urban influences are recognised, not only in the production of the heat island, but in other climate effects such as: urban rainfall, wind circulation, number of fog days, relative humidity, etc. [27,28]. These have impact not only human health, but also materials and the management of built heritage. Although Tokyo has a strong heat island, there are potentials to mitigate this, through the urban greening of buildings or increasing the area of open water in central Tokyo [29,30].

Temperature increases and a decline in relative humidity alter the potential for mould growth and change insect life cycles. The total number of hours with above 30 °C are shown in Figure 4. This will affect comfort within naturally ventilated buildings and put more pressure on the cooling systems in museum environments. It can also affect mould growth when accompanied by humid conditions, and will favour insect infestations. In Japan, the overwintering survival of insects has increased along with earlier appearances in the spring, an increase in the number of generations each year, lengthening of the reproductive season, etc. However, insects can also be susceptible to

heat stress when temperatures increase to high values from about 28 to 32 °C [31]. The northward movement of insects through the Japanese archipelago has been widely described. The distribution range of many insects have increased, including those of dragonflies [32] and the Great Mormon butterfly, *Papilio memnon* [33]. The southern green stink bug, *Nezara viridula* has moved away from the coasts to warmer regions [34]. Looking at parallels in other cities, such as London, it is possible to see a notable increase in insects such as the webbing clothes moth, *Tineola bisselliella* [35], which may be partly due to climate change, but also related to food availability, habitat and urban activities [36]. The changes in insect populations can be modelled as a function of the future climate [37,38]. Such changes create a characteristic problem with wood, as a material, which is especially important in Japanese buildings. The extensive use of wood makes the heritage particularly vulnerable to insect infestation [39]. In Tokyo, the urban heat island may cause an increase in insect numbers and variety. Additionally, there is often a greater availability of food in cities, and insects can be transported on objects or display materials. Stag beetles are popular as pets in Japan; notably, *Dorcus* spp. is an invasive insect from East and Southeast Asia, which can be released or escape [40].

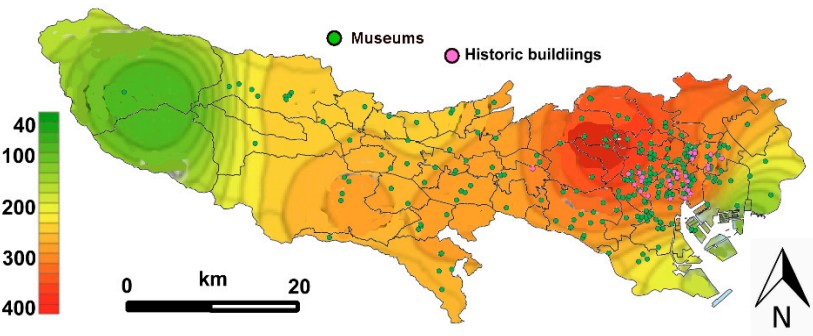

**Figure 4.** The total number of hours with temperatures above 30 °C averaged 2003–2007 over the Tokyo area. Note: Base map extracted from RealEstateJapan and JMA.

### 3.2. Storms, Typhoons and Floods

Historic floods in Tokyo are well described, for example in the great floods of 1742, the Zen monk Enjū was engaged in a pilgrimage and trapped by the intense rain, so left an account of damage to religious buildings [41]. This historic event led the Sumida River to flood, eroding the piles of the Ryōgoku Bridge, as well as damaging Eitai and Shin'ō Bridges (Figure 5a), while overflowing levees caused extensive flooding in the Kasai district [42]. The Heavy Rain Event during July 2018 [43], although not affecting Tokyo so strongly, caused tremendous damage elsewhere. There were 221 fatalities, 6296 buildings were completely destroyed, 8929 houses were inundated above the floor level, and two key buildings (National Treasures), 35 important cultural property buildings, and 24 registered cultural property buildings were affected [44].

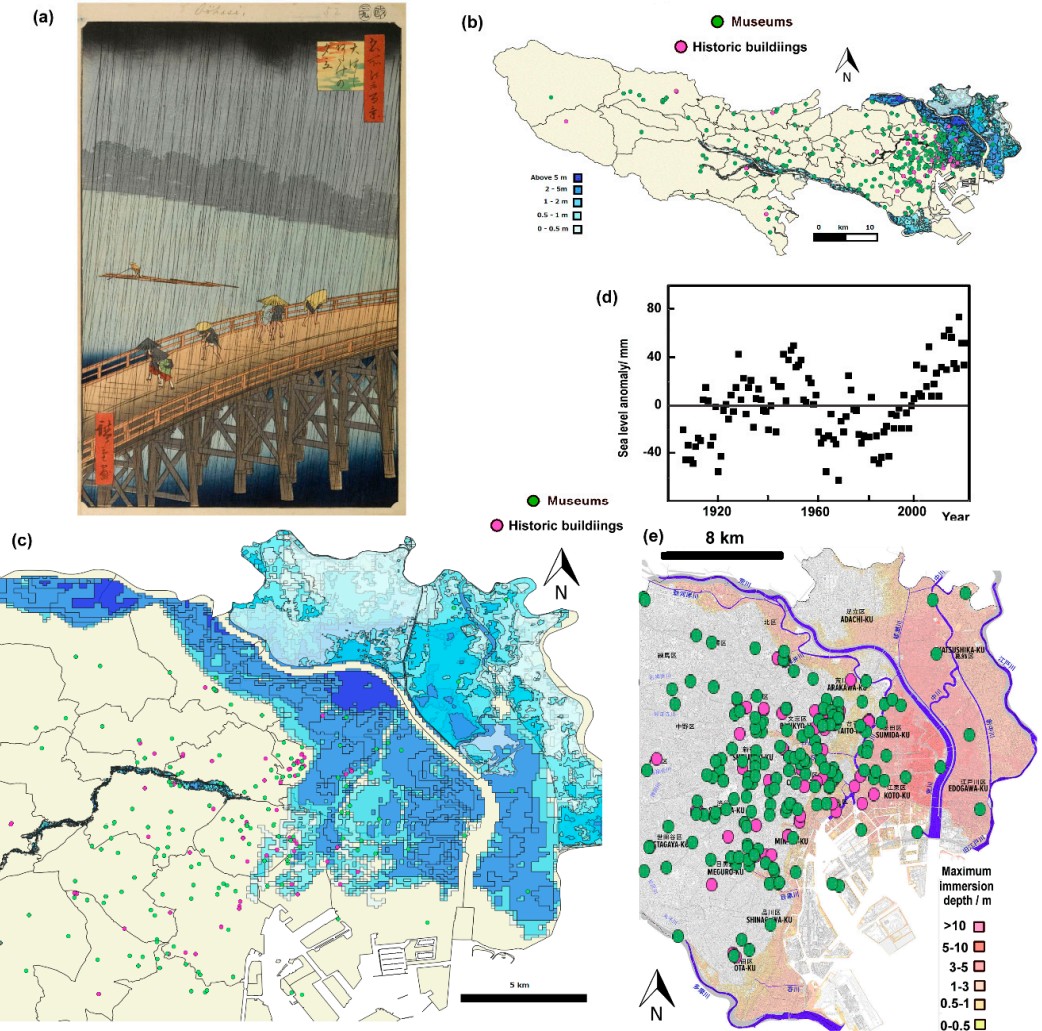

**Figure 5.** (**a**) *Sudden Shower Over Shin-Ohashi Bridge and Atake* by Hiroshige https://upload.wikimedia.org/wikipedia/commons/c/cc/Hiroshige_Atake_sous_une_averse_soudain e.jpg[45] (**b**) Area of inundation under flood conditions of Table 1. Source: (https://nlftp.mlit.go.jp/ksj/index.html [2]) (**c**) Enlargement of the central area of the inundation. (**d**) Annual mean sea level anomalies for Japan; source: Japan Meteorological Agency). (**e**) Immersion depth for the central parts of Tokyo; sources: JPC 2018, The Nikkei Shimbun, 30 March 2018, TBS News, 30 March 2018.

Changes in rainfall patterns over Japan mean that rainfall is likely to increase more than 10% over the 21st century [46]. Over the period 1976–2012, the number of hours of heavy rain (>50 mm) have increased across all regions of Japan, which is likely to continue into the future [47]. However, the picture for Tokyo is not particularly clear, although it was noted some decades ago that the urban heat island enhances convective storm activity over the city [48]. There was a period with many hours of heavy precipitation during the 1940s, though the 1990s seem to lie within the range of variation for the 20th century [49]. Nevertheless, the general view remains that there will be an increase in falls of heavy rain over the coming century [39]. In addition, there has been considerable public concern over increased frequency of stormwater flooding and its impact on residential housing [50]. Oddly, despite increases in total precipitation amount, the number of dry days with daily precipitation less than 1mm is expected to increase in almost every region of Japan. In Tokyo, a declining water table or soil moisture has the potential to expose sensitive buried archaeological sites or weaken the foundations of buildings.

The GIS mapping shows the potential risk to historic sites and museums. It reveals a high flood risk along the river and the presence of a large area of eastern Tokyo where land has

elevations below mean tide level (Figure 5b,c). Historic buildings such as the Sensō-ji Temple complex and many museums will be flooded during infrequent but large storms and, in the future, vulnerability will probably increase more than previously thought under a changing climate. Figure 5b,c show the extent of the inundation risk to museums and historic buildings. The risk is calculated adopting the conditions listed in Table 2 for each important river in the catchment. The eastern part of Tokyo can be seriously affected, with about 15% of museums at risk from high water. Some 13 museums are likely to be affected when water levels rise 2.0–5.0 m (Table 2). It is thus important to take measures to protect against flooding, especially that storage locations are required to be on upper floors, with water—and fire—proof doors. The collections in art museums are especially sensitive to damper environments. There are four art museums under threat from inundation, but three occupy galleries present on the 4th floor or above, so risk is much reduced. The Museum of Contemporary Art Tokyo and the Tokyo Metropolitan Archives are sensitive to flooding. There are 12 historic buildings in Tokyo at risk of inundation, such as Asakusa Shrine and Sensō-ji Niten-mon (Table 1), so mitigation measures are especially relevant. Longer periods of inundation can cause more damage to museums and historic buildings through a range of secondary hazards, such as mould growth and insect attack. Saraswat et al. [51] present an overview of stormwater runoff management in Tokyo to guide optimal measures and management policies within the city's governance. An increase in the future sea level (Figure 5d) would also make Tokyo vulnerable and may cause sea defences in Tokyo Bay to fail by the end of the 21st century, with increased typhoon intensity adding further threat [13].

**Table 1.** Design storms (i.e., a hypothetical discrete rainstorm) characterized by a specific duration in each river. Note full details in MLIT 2020.

| Target River | Design Storm | Rainfall | Cities Affected |
|---|---|---|---|
| Shibakawa River, New Shibakawa River | Accounts for current development of river and drainage channels, regulating ponds, etc. Rise of Shibakawa River after heavy rain that occurs every 100 years. Inundation from overflow and collapse of levees. | September 1958 flood, *Kanogawa Typhoon* or *Typhoon Ida*: 2 days rain 411 mm | Adachi, Katsushika |
| Kandagawa River | Inundation potential from the Kandagawa River due to the Tokai heavy rain accounting for channel maintenance. Depth from both river water and inside the levee. | September 2000 *Tokai heavy rainfall:* rain 589 mm, hourly maximum 114 mm | Chiyoda, Chuo, Shinjuku, Bunkyo, Taito, Shibuya, Nakano, Suginami, Toshima, Musashino, Mitaka |
| Tonegawa River | Account for current state of river channel and flood control. Simulates expectation from heavy rain once in 200 years. | Tonegawa River basin and upper Yattajima: 3 days rain 318 mm | Adachi, Katsushika, Edogawa |
| Edogawa River | Simulates Edogawa River overflow due to heavy rain that occurs about once every 200 years. | Upper Yattajima: 3 days rain 318 mm | Adachi, Katsushika, Edogawa |
| Arakawa River | Accounts for current state of the Arakawa River river channel and flood control. Simulates Arakawa | Arakawa basin; 3 days rain 548 mm | Chiyoda, Chuo, Minato, Taito, Sumida, Koto, Kita, Arakawa, Itabashi, |

|  | River overflows due to heavy rain every 200 years. |  | Adachi, Katsushika, Edogawa |
| --- | --- | --- | --- |
| Nakagawa River, Ayasegawa River | Accounts for current state of channel and flood control. Simulates Nakagawa and Ayasegawa River overflows due to 100-year heavy rain, but only inundation from Edogawa River. | Nakagawa and Ayase basins: 48 h rain 355 mm | Adachi, Katsushika |
| Tamagawa River, Oogurigawa River | Accounts for current state of channel and flood control. Simulates Tamagawa River overflow due to a 200-year heavy rain. | Tamagawa River basin and the upstream area of Ishihara site: 2 days rain 457 mm | Ota, Setagaya, Hachioji, Tachikawa, Ome, Fuchu, Akishima, Chofu, Hino, Kunitachi, Fukuo, Komae, Tama, Inagi, Hamura, Akiruno |
| Asakawa River | Accounts for development of river channels. Simulates Asakawa River overflows due to a 200-year heavy rain. | Tamagawa River basin and the upstream area of Ishihara site: 2 days rain 457 mm | Hachioji, Hino, Tama |

**Table 2.** Elevation of museums and sites in flood risk areas.

| Elevation/m | Museums | Historic Buildings |
| --- | --- | --- |
| 0–0.5 | 4 | 3 |
| 0.5–1.0 | 10 | 1 |
| 1.0–2.0 | 8 | 2 |
| 2.0–5.0 | 13 | 5 |
| >5.0 | 0 | 1 |

Modelling suggests that the frequency of typhoon landfalls will decrease (Figure 6a) and the mean value of the typhoon central atmospheric pressure will not change significantly. An important point is that the arrival probability of stronger typhoons will increase (bottom right Figure 6b) under future climate scenarios [52]. This means that flooding is a likely outcome of such increasingly intense falls of rain. Wind speeds in Tokyo in the future (2075–2099) are compared with those of the recent past (1979–2003) in Figure 6b, suggesting that the probabilities of the occurrence of higher annual maximum wind speeds will increase (i.e., exceeding 30 m s$^{-1}$), while medians of the annual maximum wind speeds decrease [53]. Figure 6c shows the relative typhoon wind risk in Japan, as the number of buildings likely to be damaged each year under the current and projected future climates. The decrease in frequency is associated with a decline in relevant typhoon events in Japan and although there is an increase in typhoon intensity, it is not enough to compensate for this decrease [53]. Dangers of wind damage in typhoons across East Asia have recently been examined along with factors that affect the intensity of damage and its extent, along with the potential to mitigate future impact [54].

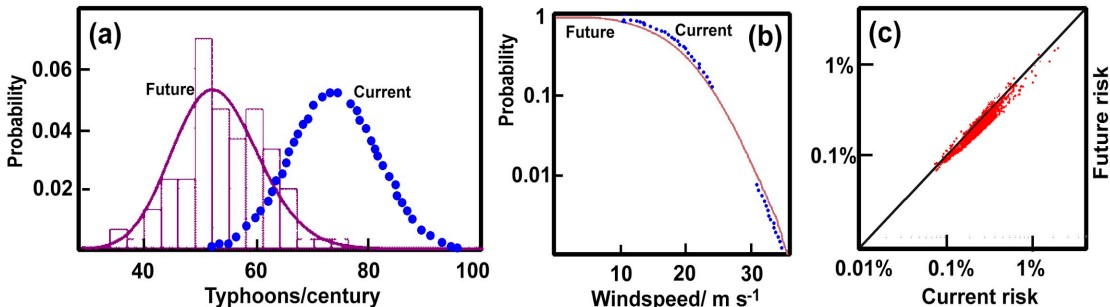

**Figure 6.** (**a**) Probable number of typhoons in Tokyo in 100 years from the stochastic typhoon model. Dotted and solid lines indicate present and future projections, respectively; see also [13]. (**b**) Exceedance probabilities for annual maximum wind speeds in Tokyo under current and projected climate. (**c**) Percentage wing risk for residential buildings in Japan under current and future scenarios [53].

### 3.3. Earthquakes and Fires

Japan is frequently affected by devastating earthquakes, because of the active seismicity caused by the subduction of the Philippine Sea Plate beneath the continental Eurasian Plate and Okinawa Plate, as well as dense distribution of inland active faults. There were major earthquakes in Tokyo in 1703 (Genroku earthquake), 1855 (Ansei Edo Earthquake) and 1894 (Meiji Tokyo Earthquake) [55], but it is the Great Kantō Earthquake of 1923 that has been especially influential (Figure 7a). It was so powerful that the 100-ton Great Buddha statue in Kamakura moved almost 60 cm. In the metropolis, the earthquakes often led to fires if they struck at lunchtime, when many people were cooking meals. Today, fires are more often caused by the rupture of gas pipelines and collapse of electricity pylons. However, at the same time, energizing fire is typical on the occasion of earthquake as it happened in the Great Hanshin-Awaji Earthquake in 1995. Extensive coastal reclamation has raised the risk of soil liquefaction during earthquakes, which are so frequent in Japan. Additionally, soil moisture and the water table are likely to affect the supporting structures of older buildings.

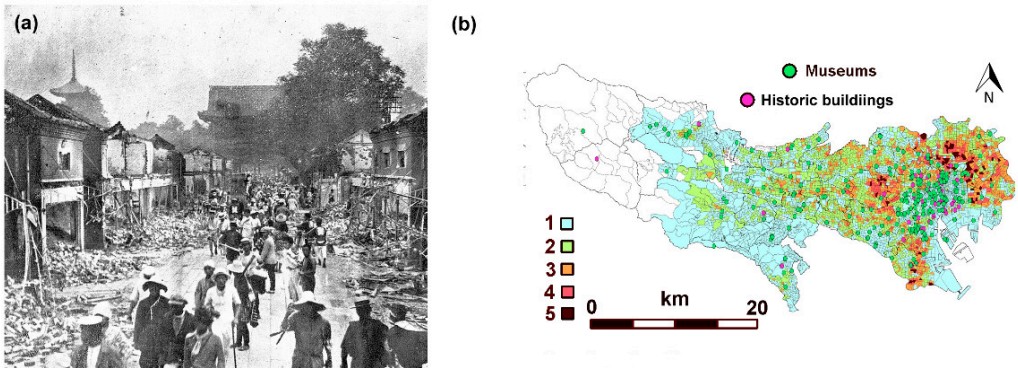

**Figure 7.** (**a**) Photograph of devastation in the area around Sensō-ji Temple, Asakusa after the Great Kantō earthquake of 1923, with smoke from fires in the background. Source: Public Domain File: Theosakamainichi-earthquakepictorialedition-1923-page9-crop.jpg [56] (**b**) The map of combined earthquake risks for Tokyo. Source: TMG 2018 and National Land Numerical Information (https://nlftp.mlit.go.jp/[2]).

Figure 7b maps the potential risk to buildings from earthquakes [57]. Although it is not tuned to heritage related structures, these sites are marked on the combined risk map of Tokyo. This combination brings together the risk of building collapse with the risk of fire. Older buildings tend

to be found in the historic downtown of Tokyo. Those along the Arakawa and Sumidagawa rivers offer lower earthquake resistance, as the structures typically have wooden or light-gauge steel frames. Machiya, Arakawa-ku is one of the worst affected areas in Tokyo, a district of densely packed wooden houses and narrow alleys, with little access for fire trucks. High risk areas are thus characterised by tightly packed buildings. Additionally, shaking can be amplified by ground characteristics, and valley and alluvial lowlands may lead to a higher risk. Fire risk is often enhanced in residential areas because of the wide use of open-flame appliances, and high density adds to the risk of fire spreading as these communities have fewer open spaces, such as parks and wide roads, which might act as fire breaks. Overall, such conditions are typical of historic areas where there is likely to be a high concentration of close-set older wooden structures (as mapped for central Tokyo in Figure 8a), with low resistance to fire [58]. Thus, the vulnerability of built heritage is not surprising, because it is most frequently associated with the older historic districts. Tokyo has been subjected to many fires. The most notable one is the Great Fire of Meireki, which occurred in 1657. It is rumoured that the fire was started when a cursed *kimono* was set on fire on a very windy day [59]. The fire spread quickly through the city, because of strong winds from the northwest, which illustrates the role of climate conditions in large scale fires. The conflagration that followed the Great Kantō Earthquake destroyed neighbourhoods and some significant sites, such as the Metropolitan Police Department building (Figure 8b). Such characteristics of central Tokyo contributed to the enormous loss of historic buildings and residences from fires after incendiary bomb raids during World War II.

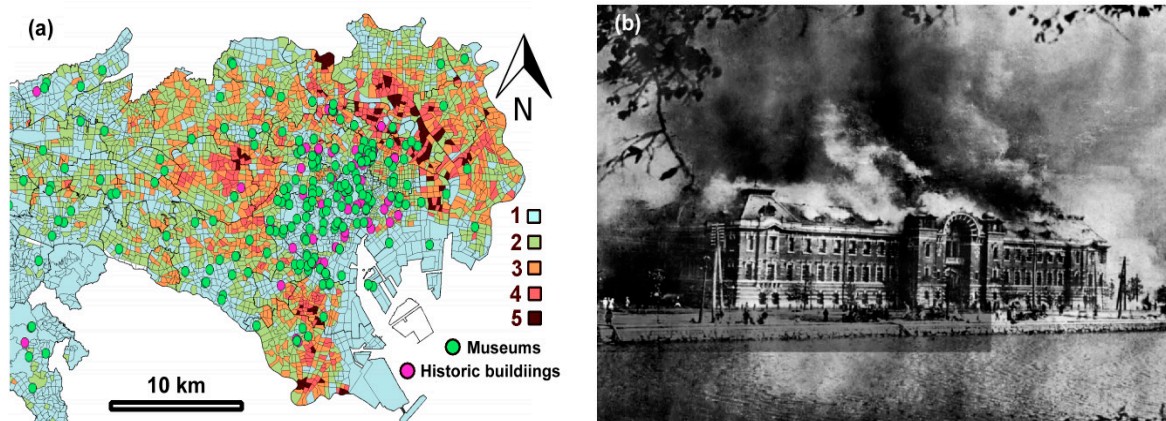

**Figure 8.** (**a**) Fire risk for Tokyo. Source: TMG 2018 and National Land Numerical Information (https://nlftp.mlit.go.jp/). (**b**) The Metropolitan Police Department building burning after the Great Kantō Earthquake. Source: Public Domain—File: Metropolitan Police Office after Kanto Earthquake.jpg [60]

There is the potential for increased risk of fire both after earthquakes and that induced by lightning strikes. Shindo et al. [61] show that the number of lightning flashes per year over Japan and has increased over the period 1992 to 2008. Additionally, Tokyo is likely to see enhanced convective storm activity in future, so an increased frequency of lightning strikes and fires might be expected. The potential for climate change to lead to more lightning strikes can be countered by improved protection against lightning and incorporating fire-suppression devices in buildings. The fire at Notre-Dame de Paris on the evening of 15 April 2019 initiated many concerns in Japan, yet soon after, there was a devastating fire at Okinawa's Shuri Castle in 31 October 2019. It was followed almost immediately by a fire on small thatched-roof huts at a car park in Ogi-machi, one of the   components of a World Heritage property named Historic Villages of Shirakawa-go and Gokayama in Gifu Prefecture. Wooden structures and thatched roofs are especially vulnerable, so have been of great concern for many hundreds of years, so Japan developed a range of approaches to limit the extent of damage from fire [39]. These fires in 2019 encouraged expanded budgets for heritage protection in Japan, but also more modest activities; e.g., the Tokyo Fire Department offers

fire prevention guidance to Zōjō-ji temple, a complex which has survived many fires since the early 17th century, even though many of its components, such as the mausoleums of the Tokugawa Shoguns, were burnt down during raids of World War II.

*3.4. Debris Flow, Slope Failure and Landslides*

A range of sediment disasters are induced by heavy rain, but also triggered by earthquakes. These can cause catastrophic damage to built heritage, and such sediment flows include:

1. Debris flow—soil and rock on a hillside or in a riverbed are washed after heavy or continuous rain, which can reach 20–40 km hr$^{-1}$
2. Slope failure—abrupt slope collapse under the influence of a rain or an earthquake. This occurs so suddenly that people fail to escape when it occurs near a residential area and can lead to many fatalities.
3. Landslide—massive quantities of soil move slowly downslope under the influence of groundwater and gravity. The large soil mass means serious damage can occur, and once started, it is extremely difficult to stop.

Sediment hazards are more probable in the western parts of Tokyo, because of the mountainous geography and shown in Figure 9 which shows the red and yellow risk areas.

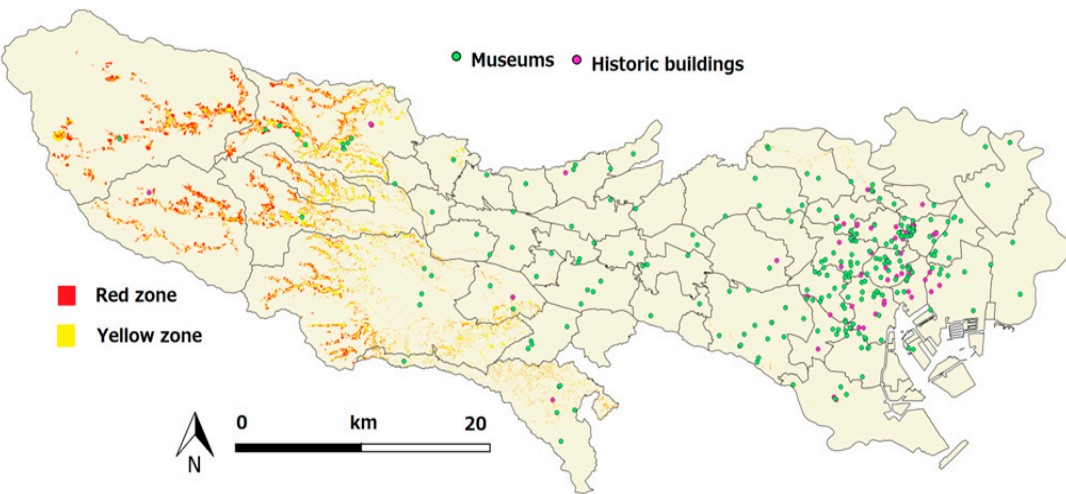

**Figure 9.** Sediment disaster map of the Tokyo area. *Red Zone* where slopes can collapse and damage buildings and threaten lives. *Yellow Zone* where steep slopes can collapse, risking injury or life. Source: National Land Numerical Information (https://nlftp.mlit.go.jp/[2]).

*Red Zone*: Where steep slopes are likely to collapse, buildings are damaged; areas where there is a risk of significant damage to the lives or physical health of residents. In Japan, a permit is required for specific development activities and structural regulation for buildings.
*Yellow Zone*: Where steep slopes are likely to collapse, risking injury or life, and when the danger is apparent, a warning and evacuation system is to be in place.

Red and yellow zones are more predominant in western Tokyo, but there are many popular places in the eastern and central districts such as the NHK Museum of Broadcasting and the Itabashi Art Museum, which even though not in red or yellow zones, there may be some risk as the sites are close to potential sediment threats.

There are nine museums in the yellow zone: seven are in the western area and two in the eastern part of Tokyo. Additionally, there are two historic buildings in yellow zone: one is in the

western area and the other in the eastern part of Tokyo. The west has fewer historic buildings and museums than the east, but still there are many small red and yellow zones liable to sediment disaster in central Tokyo. This means that the risk is not only to the western part, but also the eastern, where there are relatively fewer sites likely to be threatened. In the west, which is more mountainous, the yellow zone is occupied by: Okutama Water and Green Friendship Hall, Gyokudo Art Museum, Ome Kimono Museum, Yoshikawa Eiji Museum, Ome Municipal Museum of Provincial History, Akiruno City Itsukaichi Museum and the Hamurashi Kyodo Museum. Protection can best take the form of slope maintenance, well-practiced in Hong Kong [62], which has steep terrain and receives heavy falls of rain, especially during typhoons.

### 3.5. Visitor Experience Under a Changing Climate

As shown in Figure 3, many external parameters affect visitor choices. Typically, wet weather can discourage travelling round the city, but on the other hand it can mean a greater tendency to go indoors to shelter [63]. Many individuals appear to actively adjust their plans throughout the day in response to rain. However, for others, attendance depends upon prior weather forecasts of rain. The duration of a visit is also likely to increase during rainy periods [64]. However, in the hotter weather of the future, visitors may want to escape from the oppressive outdoor heat, especially to air-conditioned museum interiors. In historic dwellings where mechanical ventilation is not appropriate, the heat of the interiors may be such that visitors will be driven outside and away from the poor air movement among crowds of visitors. The change in visitor behaviour is obviously not simply a matter of the changing climate and must also consider changes to visitor types and different patterns of behaviour and expectations.

Museums and historic buildings are increasingly broadening the range of visitors they attract, so it is no longer the well-informed tourist, with guidebook in hand. Plans, for example at the UK's National Trust, aim to welcome a wider variety of visitor types they categorise as: (i) active thinkers, (ii) live life to the full, (iii) spontaneous characters, (iv) young experience seekers and, among groups with children, (v) the family group, (vi) kids first families, and (vii) home and family [65]. Encouraging these new groups of visitors will inevitably change the way the heritage sites are experienced and used. We can imagine that the gentle tours through historic rooms may be less important than more active pursuits and greater use of the grounds and surroundings.

In recent years, the sunnier urban environments have led to more oxidizing pollutants and meant that the surfaces of buildings have taken on a somewhat warmer tone [66]. In addition to chemical drivers, shifting colours may be a product of biological growth. As the climate changes, viewing the beauty of a snowy Tokyo, reminiscent of the paintings of Hiroshige, is likely to be more difficult (Figure 10). The changing seasons have influenced the viewing of cherry blossoms (*sakura*) and the autumn leaves, the dates of these activities have had to shift as the warm seasons have grown longer [67].

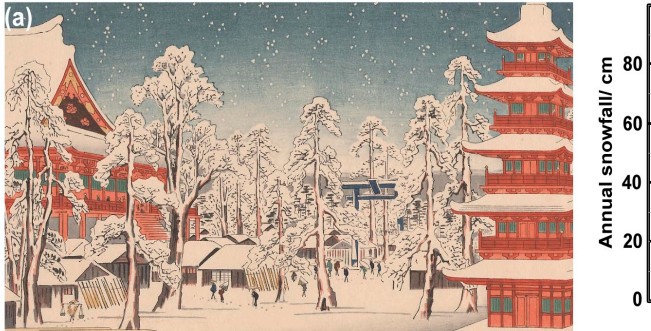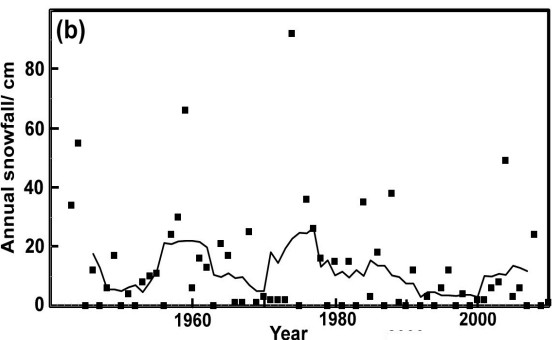

**Figure 10.** (**a**) Hiroshige's *Snow Scene at Kinryuzan Buddist Temple, Asakusa District* (**b**) Annual Tokyo snowfall. The line shows the 7-year running mean. Note: Image is part of *Toto Yukimi Hakkei* (8 Snow Scenes of the Eastern Metropolis): http://mercury.lcs.mit.edu/~jnc/prints/8snow.html [68]. Data from WMO Station ID:47662 of JMA.

## 4. Discussion and Conclusions

This study presents maps of some natural pressures on cultural heritage in Tokyo. Accardo et al. [4] claim maps of heritage risk to be useful, because they provide a "deeper and more concise knowledge of risk intensity than may be obtained by comparing the potential danger level of any single cultural property, based on the three static/structural, human impact, and environmental factors, with its state of conservation (vulnerability index) recorded". It is easy to accept such thoughts, but the translation of maps into management policies is not trivial. Maps of potential risk are strategic, so they seem more useful to large organisations or government agencies. Such entities need to consider the broader picture and prioritise and compare risks over an extensive range of sites. Ultimately, they need to allocate limited resources among sites with competing, but necessary demands; and budgets are always tight. It is also possible that the maps can be of some value to a single museum or site, as they may hint at risks of which they are unaware, but often they will know many of the risks that confront them. In Italy, Istituto Superiore per la Conservazione ed il Restauro (High Institute for Conservation and Restoration) has developed a risk map named Carta del Rischio del Patrimonio Culturale (Risk map of Cultural Heritage) since the 1990s. It integrates the national inventory of cultural heritage and storage building of artistic objects such as museums, and hazard maps of natural/human-induced disasters, to utilise it for establishing restoration plans of cultural heritage by Soprintendenze, national authorities of cultural heritage in the regions of Italy.

Risk maps can reveal the changes in climate and pressures from natural disasters. They can be open and provide information to stakeholders and the public. The risks may become evident to more individuals and encourage risk reduction. While it is possible to create overlapping maps with GIS, the availability of data is often limited with only a few of the desirable maps are available. For instance, Tokyo tsunami data have yet to be made available in a useful format, although the municipality webpage gives tsunami hazard maps. In a publicly available GIS form, they can more readily be used to identify and visualise built heritage hazards and sites at risk in an effective way.

Risk management and assessment in general is relatively advanced in Japan, so there is a hazard map portal site (https://disaportal.gsi.go.jp/index.html [69]). In contrast, application of risk maps to cultural heritage protection from natural/human-induced disasters is not so frequent. National-scale risk maps are useful for policy making such as allocation of resources, but smaller-scale maps dealing with multiple disasters should be necessary for site managers and museum curators who take care of one or small numbers of site or museums. Up to now, disaster preparedness planning has used such material for establishing evacuation routes for visitors and objects, but less for planning mitigation strategies. Risk maps can also be useful in designing new museums by assessing risks in advance. A general problem in creating risk maps is that most of the publicly available data focusses on general built structures rather than built heritage, so it would be best to develop techniques that might tune it for that use. Historic buildings, especially, are more sensitive to hazards. In this study, museums and historic buildings designated as national treasures and important cultural properties have been considered, but it is also characteristic of large cities like Tokyo that there are many privately owned cultural properties, and they should also be stored safely. There are other areas worthy of future expansion, e.g., much of our knowledge of insects is related to those which are agricultural pests, rather than those that place heritage at risk. A better understanding of the life cycles of wood-boring insects in Japan would be useful. An understanding of natural hazards is well developed in Japan, and the science of climate change is nicely represented by the Japan Meteorological Agency, but there is only limited effort to tune this to a future heritage climatology. Similarly, a knowledge of air pollution, both indoors and out, has largely been concerned with the impact on human health. The example of Tokyo can be applied to other large cities, which often have rivers and similar characteristics, so some risks are shared. Cultural heritage is affected by climate change, as this alters the nature of many hazards. It is evident that more work is necessary to improve the health of our heritage.

**Author Contributions:** "Conceptualization, P.B. and M.H. and Y.F.; methodology, M.H. and Y.F.; data analysis P.B; investigation and writing P.B. and M.H. All authors have read and agreed to the published version of the manuscript.

**Funding:** This research received no external funding.

**Acknowledgments:** We would like to thank Helen Lloyd of the UK National Trust for her usual thoughtful advice.

**Conflicts of Interest:** The authors declare no conflict of interest.

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
