# Peer review of "Mapping Climate Change, Natural Hazards and Tokyo’s Built Heritage"

_atmosphere, doi:10.3390/atmos11070680_

Round 1

Reviewer 1 Report

General:  A short summary on each paragraph describing certain risks could be helpful.

Editorial comments:

Title:

Please check the Affiliation of Author 1 : “National Sun Yat-Sen University, Kaohsiung,” is doubled

Abstract:

Comma missing? ”as a response to environmental issues, from SARS”…

Add word?  “pipelines and collapse of “electricity” pylons”…

Page 3:

Use “,” instead of “.” – “for other purposes to heritage. e.g. seismic”…

Page 4:

Fig. 3: Please revise Y-Axis – really only 5 to up to 30 international visitors?

Please revise: “Globally, the temperature is likely to increase by about 0.8 °C through the 20th century.“

“Projecting the regional climate of Eastern Japan over the periods 1980–1999 and 2076–2095 suggests that the annual mean temperature will increase by more than 3 °C.” Compared to what? Average of the baseline? Etc.???

Page 5:

Please revise (remove “but”) : “Although Tokyo has a strong heat island, but there are potentials to mitigate this through urban greening of buildings or increasing the area of open water in central Tokyo (Ichinose et al., 1999; Mochida et al., 1997).”

Page 6:

Spelling: “35 important heritage buildingsn“

Page 9: Caption wrong, please revise: “Table 1” or rather swap both tables in the article (Table 2 currently before Table 2).

Please explain in the text the concept of a “Designed storm”. What is meant?

Page 14:

Consider revising: “We also have to consider the change in the visitor the type of visitor and the change in their behaviour and expectations in terms of visits to historic sites.”

Please give a source to the following: “UK’s National Trust aim to welcome a wider variety of visitor types”

please remove “.”: “As the climate changes. viewing the beauty”

Author Response

Referee 1

Page 4:

Fig. 3: Please revise Y-Axis – really only 5 to up to 30 international visitors?
INDEED THE MISSING MILLIONS IS ADDED!

Please revise: “Globally, the temperature is likely to increase by about 0.8 °C through the 20th century.“
YES CORRECTED
“Projecting the regional climate of Eastern Japan over the periods 1980–1999 and 2076–2095 suggests that the annual mean temperature will increase by more than 3 °C.” Compared to what? Average of the baseline? Etc.???
IT IS BETWEEN THE TWO PERIODS MENTIONED. THIS IS N OW MADE CLEARER

Page 5:
Please revise (remove “but”) : “Although Tokyo has a strong heat island, but there are potentials to mitigate this through urban greening of buildings or increasing the area of open water in central Tokyo (Ichinose et al., 1999; Mochida et al., 1997).”
DONE

Page 6:
Spelling: “35 important heritage buildingsn“
DONE

Page 9: Caption wrong, please revise: “Table 1” or rather swap both tables in the article (Table 2 currently before Table 2).
Please explain in the text the concept of a “Designed storm”. What is meant?
DONE AND DEFINED IN TABLE HEADING "Table 2. Design storms (i.e. a hypothetical discrete rainstorm) characterized by a specific duration in each river. Note full details in MLIT 2020"

Page 14:
Consider revising: “We also have to consider the change in the visitor the type of visitor and the change in their behaviour and expectations in terms of visits to historic sites.”
Please give a source to the following: “UK’s National Trust aim to welcome a wider variety of visitor types”
REFERENCE 61 ADDED

please remove “.”: “As the climate changes. viewing the beauty”
DONE

Reviewer 2 Report

Dear authors,

 please rewrite abstract to prevent plagiarism.

Figure 1: please add north arrow, the title for this map. Also please add while Line in the legend.

Figure 2: Do you get permission for that map? I would recommend you to draw it by your team.

Figure 4: Please add the title in this map

Figure 5e: Please provide permission for this one.

Figure 7a, Figure 8b: Do you have permission to use those photos?

Figure 9: I recommend this figure should draw by your team, and add title in this map

The paper doesnt have Line, so it is hard to indicate that which line need to rewrite, therefore please check this paper I did not which sentences need to be rewritten. 

The paper did not follow journal format, please check citation through this paper. 

References are wrong format.

Author Response

Dear authors,

please rewrite abstract to prevent plagiarism.
WE ARE PUZZLED BY THIS, BUT DID NOT HAVE ANY ENGLISH LANGUAGE PLAGIARISM CHECKERS SO HOPE ANY PLIAGIARISM AS ACCIDENTAL

Figure 1: please add north arrow, the title for this map. Also please add while Line in the legend.
NORTH HAS BEEN DELETED FROM THE CAPTIONS AND ADDED AS AN ARROW IN THE DIAGRAMS
Figure 2: Do you get permission for that map? I would recommend you to draw it by your team.
THIS MAP IS DRAWN BY THE GIS SOFTWARE SO IN A WAY IS OURS
Figure 4: Please add the title in this map
ADDED "over the Tokyo area."IN THE CAPTION
Figure 5e: Please provide permission for this one.
REFERENCED AS "2018, The Nikkei Shimbun, March 30, 2018, TBS News, March 30, 2018. "
Figure 7a,
ADDED Public Domain File:Theosakamainichi-earthquakepictorialedition-1923-page9-crop.jpg
Figure 8b: Do you have permission to use those photos?
ADDED Public Domain - File:Metropolitan Police Office after Kanto Earthquake.jpg
Figure 9: I recommend this figure should draw by your team, and add title in this map
THIS MAP IS DRAWN BY THE GIS SOFTWARE SO IN A WAY IS OURS. ADDED "over the Tokyo area."IN THE CAPTION

The paper doesnt have Line, so it is hard to indicate that which line need to rewrite, therefore please check this paper I did not which sentences need to be rewritten.

The paper did not follow journal format, please check citation through this paper.
References are wrong format.
REFERENCE FORMAT IS CHANGED THROUGHOUT

Reviewer 3 Report

Dear Authors,

the manuscript is arranged in a very clear way and based upon the best practice in the field. You were effective in highlighting the peculiarities of the specific urban context and this certainly results in a strength of your research.  

Cross fingers.

Best regards.

Author Response

NO CHANGES SEEM TO BE IMPLIED, BUT WE THGAN THE REFEREE FOR THEIR SUPPORT 

Yes Can be improved Must be improved Not applicable
Does the introduction provide sufficient background and include all relevant references?
(x) ( ) ( ) ( )
Is the research design appropriate?
(x) ( ) ( ) ( )
Are the methods adequately described?
(x) ( ) ( ) ( )
Are the results clearly presented?
(x) ( ) ( ) ( )
Are the conclusions supported by the results?
(x) ( ) ( ) ( )
Comments and Suggestions for Authors
Dear Authors,

the manuscript is arranged in a very clear way and based upon the best practice in the field. You were effective in highlighting the peculiarities of the specific urban context and this certainly results in a strength of your research.

Round 2

Reviewer 2 Report

The authors have answered all my comments. Good job.